# Real-Time CLAHE Algorithm Implementation in SoC FPGA Device for 4K UHD Video Stream

**Tomasz Kryjak \***[ID]**, Krzysztof Blachut**[ID]**, Hubert Szolc**[ID]** and Mateusz Wasala**[ID]

Embedded Vision Systems Group, Computer Vision Laboratory, Department of Automatic Control and Robotics, AGH University of Science and Technology, Al. Mickiewicza 30, 30-059 Krakow, Poland; kblachut@agh.edu.pl (K.B.); szolc@agh.edu.pl (H.S.); mateusz.wasala@agh.edu.pl (M.W.)
**\*** Correspondence: tomasz.kryjak@agh.edu.pl

**Abstract:** One of the problems encountered in the field of computer vision and video data analysis is the extraction of information from low-contrast images. This problem can be addressed in several ways, including the use of histogram equalisation algorithms. In this work, a method designed for this purpose—the Contrast-Limited Adaptive Histogram Equalization (CLAHE) algorithm—is implemented in hardware. An FPGA platform is used for this purpose due to the ability to run parallel computations and very low power consumption. To enable the processing of a 4K resolution (UHD, 3840 × 2160 pixels) video stream at 60 fps (frames per second) by using the CLAHE method, it is necessary to use a vector data format and process multiple pixels simultaneously. The algorithm realised in this work can be a component of a larger vision system, such as in autonomous vehicles or drones, but it can also support the analysis of underwater, thermal, or medical images both by humans and in an automated system.

**Keywords:** CLAHE; Contrast-Limited Adaptive Histogram Equalization; 4K; UHD; FPGA; SoC; Zynq UltraScale+; real-time image processing; histogram equalization



## 1. Introduction

Image processing and analysis is currently an element of many automatic systems. With the information obtained from vision sensors, many machines, robots, or autonomous vehicles can operate correctly. In order to facilitate the extraction of relevant information from an image, pre-processing methods are often used to improve its quality in some way before the target algorithm is applied. Such operations include various types of filtering (blurring, de-noising), colour space conversions, or contrast enhancement [1]. Many images are characterised by low contrast, e.g., due to the conditions in which their acquisition is performed. Typical examples are thermal imaging, underwater images, and also the automotive industry (e.g., image enhancement of cameras supporting the view from car mirrors), or medical imaging—e.g., X-ray or computed tomography (CT). In the case of the latter two categories, image-contrast enhancement can make it easier for people to make the right decisions (e.g., for drivers to change lanes, for doctors to make an effective diagnosis), so it is an important component not only for fully automatic systems. Furthermore, it is worth noting that this type of pre-processing can also be applied to solutions with deep convolutional neural networks.

One of the widely used methods to improve image quality is histogram equalisation. The basic algorithm is known as the Global Histogram Equalization (GHE) (in names we use the US spelling, as more popular in the scientific world) and consists of the following steps:

- determining the histogram for the entire image frame (the image is assumed to be in greyscale);
- calculating the cumulative histogram and normalising it (to the range of 0–255); and
- performing look-up table (LUT) operations on the image, with recoding in the form of a normalised cumulative histogram.

As a result, low-contrast areas are highlighted and high contrast areas are "flattened". This approach works well mainly for images with relatively homogeneous illumination. In more challenging cases, the results are usually not satisfactory.

An extension of this method is the Adaptive Histogram Equalization (AHE) algorithm [2]. In this approach, the image is divided into windows, in which the histogram equalisation process is performed. Therefore, this method is a local approach. The algorithm preserves details in heterogeneous images, but has a disadvantage of emphasising (amplifying) noise in homogeneous areas.

The Contrast Limited Adaptive Histogram Equalization (CLAHE) algorithm does not have such a disadvantage [3]. In this solution, the values of individual histogram bins are limited, and the resulting excess is redistributed among the remaining bins. The results of the three mentioned methods are illustrated in Figure 1. Section 2 describes in detail the subsequent steps of the CLAHE algorithm.

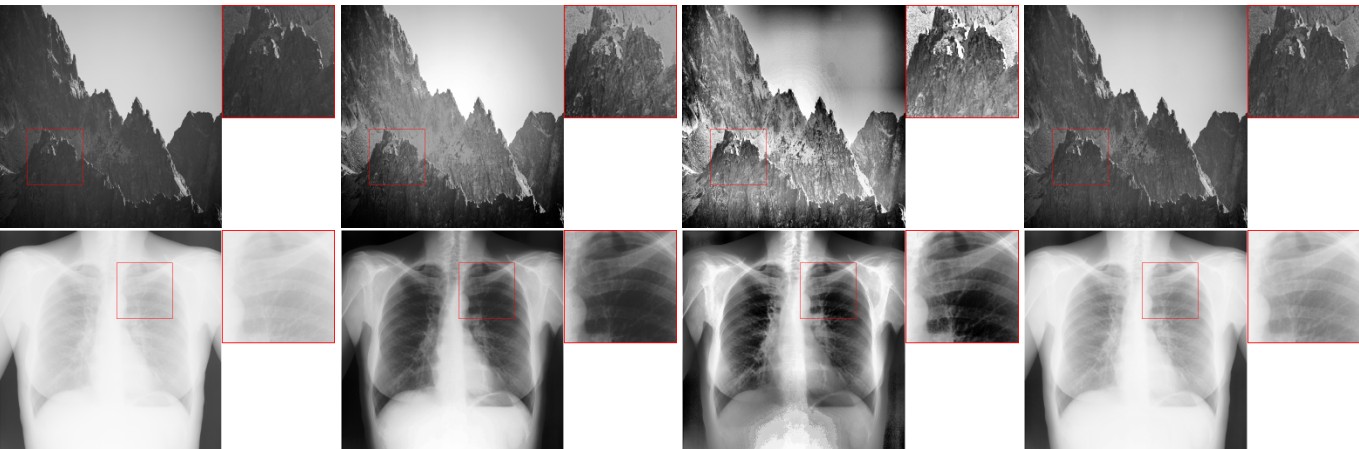

**Figure 1.** Comparison of Global Histogram Equalization (GHE), Adaptive Histogram Equalization (AHE) and Contrast Limited Adaptive Histogram Equalization (CLAHE). The figures show successively (from the left) the original image, the image after applying GHE, AHE, and CLAHE algorithms. The GHE algorithm does not perform well when there is a significant difference between the highest and the lowest intensity in the image. The main drawback of the AHE is visible in the homogeneous regions (like sky in the mountains image). The CLAHE algorithm performs well in both situations.

For some of the mentioned applications (e.g., from the automotive and robotics industry), it is desirable that histogram equalisation algorithms operate in real-time for images of the highest possible quality. This is a demanding task, especially for more complex algorithms. It is even more demanding in the case of processing a video stream, understood as many consecutive images (30–60 frames per second as standard). In a general case, this is difficult to achieve by using a CPU. It is therefore necessary to use hardware platforms that enable parallel and pipeline data processing. These include graphics processing unit (GPU), embedded GPU, field programmable gate array (FPGA), or other dedicated hardware platforms (e.g., application-specific integrated circuit (ASIC)). The unquestionable advantages of FPGAs are energy efficiency, the possibility of operation without an additional host (as in the case of GPUs), and high flexibility, contrary to ASICs.

FPGAs have found applications in many real-time vision systems. They are used for optical flow determination, e.g., with the Lucas–Kanade and Horn–Schunck methods [4], or stereo correspondence with the Semi-Global Matching algorithm [5]. They also enable the implementation of advanced object tracking methods [6]. FPGAs can be applied in advanced driver assistance systems, for example for high-speed gaze detection [7], and unmanned aerial vehicles, for example the simultaneous localization and mapping [8]. In the last few years, a very large number of scientific and industrial works also addresses the topic of deep neural networks acceleration (especially convolutional neural networks) on FPGAs [9].

The main contributions of this paper are as follows.

- We propose a hardware implementation of the CLAHE algorithm on an FPGA platform, enabling real-time processing of a 4K (Ultra HD) video stream, which to our best knowledge has not been done before; and
- We use a vector stream format (4 ppc) to implement the CLAHE algorithm, which should be considered as an architectural novelty due to required redesign of its components.

The remainder of this paper is organised as follows. In Section 2 we present the CLAHE algorithm with its typical components and highlight the most typical applications. We discuss the previously published work in Section 3. In Section 4 we describe our hardware implementation of the CLAHE algorithm on an FPGA platform. The results we obtained are presented in Section 5, where we also compare them with state-of-the-art solutions. In Section 6 we discuss possible modifications to the CLAHE algorithm and their hardware implementations. Section 7 summarises our work and indicates directions for further development.

## 2. CLAHE Algorithm

In general, the CLAHE algorithm can be implemented in several variants, depending on the adopted implementation assumptions. Differences may include the use of a mechanism for dividing the image into windows or allowing a slight exceedance of the set limit of the histogram value. The version of the CLAHE algorithm used by us can be divided into four stages:

- the division of the image into rectangular, non-overlapping windows;
- the computation of the histogram for each window and its redistribution;
- the calculation of the LUT mapping function; and
- the interpolation of the resulting pixel values.

An overview of the CLAHE algorithm is presented in Figure 2.

Adaptive histogram equalisation is most often performed on greyscale images or for the luminance (brightness) component of a colour image (V from HSV, Y from YCbCr, L from CIELab etc.). It is also possible to apply a given algorithm (e.g., CLAHE) to all or selected channels of a colour image, performing operations on each channel separately. However, this operation may have a significant impact on the colouring of the image. In this project, we focus on the greyscale images, although the described algorithm can be easily applied to other mentioned cases (as discussed in Section 6).

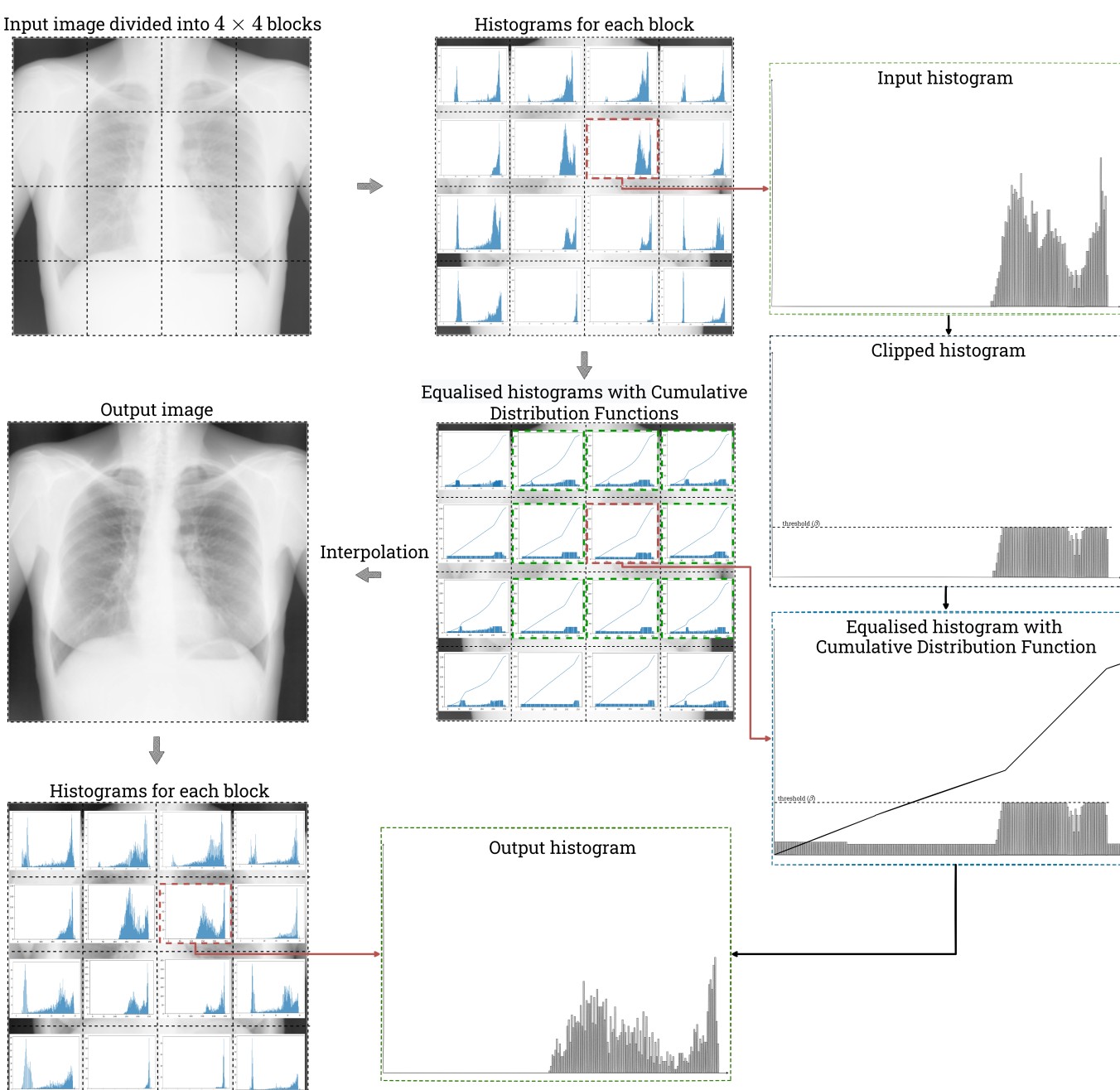

**Figure 2.** Overview of the CLAHE algorithm. First, the input image is divided into rectangular, non-overlapping windows (blocks, tiles). Second, for each of the tiles, a clipped histogram is computed. The excess is then redistributed and the cumulative distribution function (CDF) is computed. Finally, by using the CFD as the mapping function and interpolating the pixel values, the new image is obtained. Note that for each tile, the histograms are more uniform.

### 2.1. Tiles Generation

Two main solutions to the window partitioning problem are commonly used in the literature. In the first one, for each pixel, the window is independent (similar to context filtering). Then, there is no problem of visible borders between blocks—transitions between individual pixel values are smooth. However, the main disadvantages are higher computational complexity and the problem of determining the values of boundary pixels. This problem is quite important, because the size of the window in which the histogram is calculated is relatively large, e.g., $64 \times 64$ pixels. In the second approach, the image is divided into non-overlapping (i.e., disjoint) windows (rectangular or square). Then,

the aforementioned problem of visibility of boundaries between blocks occurs , but it is eliminated by interpolation. On the other hand, the boundary pixel problem does not exist. In general, hybrid variants are also possible, i.e., partially overlapping windows.

The second of the variants described is more common. In its case, three types of windows can be distinguished, as in Figure 3: CR (corner regions), BR (border regions), and IR (inner regions). The distinction between these blocks affects the interpolation performed when determining the output pixel values.

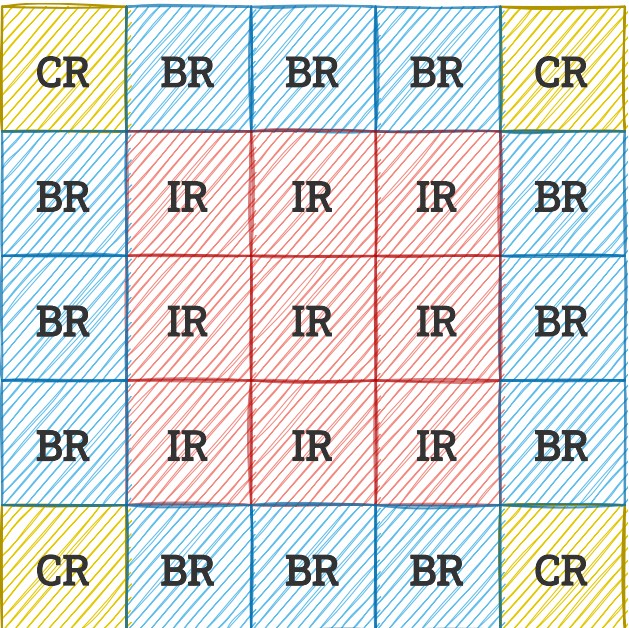

**Figure 3.** Example of division of an input image into blocks, corner regions (CR), border regions (BR), and inner regions (IR).

### 2.2. Histogram Calculation and Redistribution

In the first step of the actual algorithm, a typical histogram with 256 bins is determined for each block (sometimes variants with fewer bins are also used). It can be described by Equations (1) and (2),

$$h(n) = \sum_{i=0}^{XX-1} \sum_{j=0}^{YY-1} g(n, i, j) \text{ for } n = 0, 1, \dots, N - 1 \tag{1}$$

$$g(n, i, j) = \begin{cases} 1 & \text{if } I(i, j) = n \\ 0 & \text{otherwise} \end{cases} \tag{2}$$

where

- $n$ is the grey level, histogram bin;
- $h(n)$ is the histogram value for the $n$-th bin;
- $N$ is the number of histogram bins (256 in this case);
- $XX, YY$ are the dimensions of the image block;
- $i, j$ are the coordinates of a pixel;
- $g(n, i, j)$ is the function that determines whether the value of a pixel with coordinates $(i, j)$ is equal to $n$; and
- $I(i, j)$ is the value of the pixel with coordinates $(i, j)$.

In the second step, overabundant pixels are counted (this step and the subsequent ones do not occur in the AHE method, i.e., without contrast reduction). For this purpose, a threshold $\beta$—the maximum value that a single histogram bin can have—is defined. It should be added that usually this parameter is calculated from Equation (3).

$$\beta = b \cdot XX \cdot YY \tag{3}$$

where: $b$ is the parameter in the range 0–1 (the value 1 indicates the AHE method; it usually takes small values: 0.01, 0.02, 0.1), and $XX, YY$ is the block size into which the image is divided. Pixels exceeding this threshold are counted (this value is referred to as excess) and the histogram itself is clipped. We show this process graphically in Figure 4,

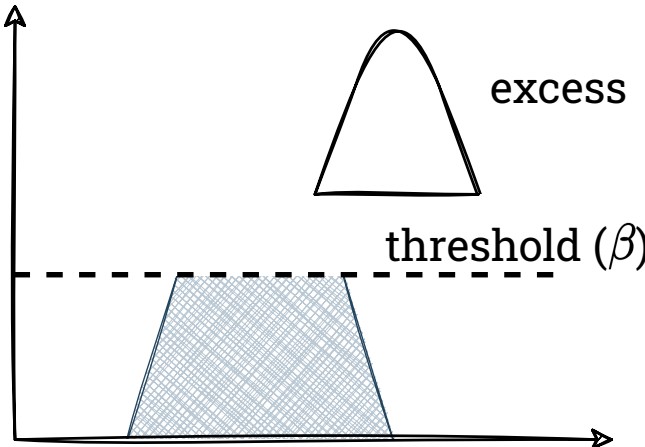

**Figure 4.** First step of the redistribution procedure: calculation of excess.

In the third step, an initial redistribution of the overabundant pixels to the remaining bins of the histogram is performed. For this purpose, an auxiliary variable $m$ is determined according to Equation (4),

$$m = \frac{excess}{N}. \tag{4}$$

This value is added to each bin for which this operation would not result in an exceedance of $\beta$. Bins for which adding $m$ would result in an exceedance of $\beta$ are only completed up to that value. We show this schematically in Figure 5.

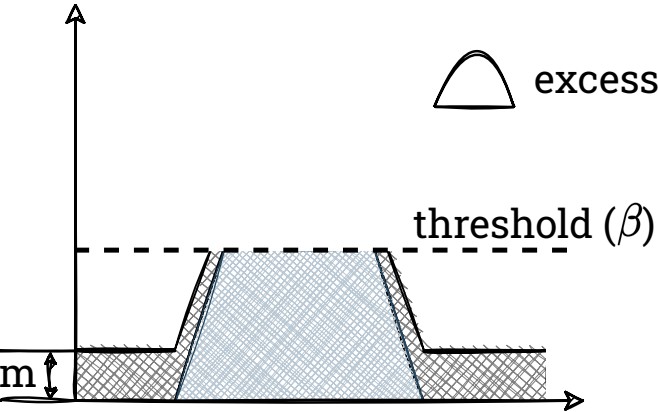

**Figure 5.** The second stage of the redistribution procedure: the first iteration of redistribution may result in some remaining excess.

In the last step, the remaining excess is evenly distributed among all allowable bins. This is usually implemented in a "while" loop, which poses a problem with hardware implementations (no pre-determined number of iterations). The final result of the procedure is shown in Figure 6.

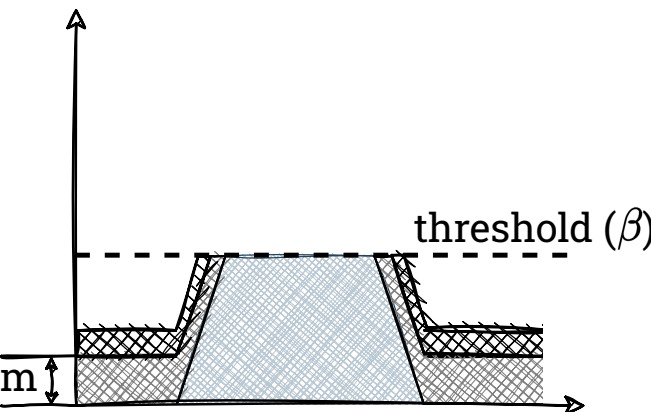

**Figure 6.** Results of the redistribution procedure: the whole excess is redistributed.

The entire redistribution process can be written in the form of pseudocode, which we present in Listing 1.

**Listing 1.** The redistribution algorithm in its basic version, described in [10].

```
excess = 0;
for (i = 0; i < N; ++i) {
  if (h[i] > β) {
    excess += h[i] − β;
  }
}

m = excess / N;
for (i = 0; i < N; ++i) {
  if (h[i] < β − m) {
    h[i] += m;
    excess −= m;
  }
  else if (h[i] < β) {
    excess += h[i] − β;
    h[i] = β;
  }
}

while (excess > 0) {
  for (i = 0; i < N; ++i) {
    if (excess > 0) {
      if (h[i] < β) {
        h[i] += 1;
        excess −= 1;
      }
    }
  }
}
```

### 2.3. Mapping Function

The mapping function is determined as in the standard HE method. First, the cumulative histogram (cumulative distribution function (CDF)) is calculated. It is then normalised (to $[0, 1]$) and finally multiplied by the maximum pixel value, i.e., 255. This process is expressed by Equation (5),

$$f_{i,j}(n) = \frac{N-1}{M} \sum_{k=0}^{n} h_{i,j}(k) \tag{5}$$

where:

- $i, j$ are the coordinates of the image window;
- $M$ is the number of pixels in the window;
- $N$ is the number of grey levels (histogram bins); and
- $h_{i,j}$ is the histogram of the image window with coordinates $(i, j)$.

### 2.4. Bilinear Interpolation

The resulting values of the distribution are ultimately used to determine the new output pixel values. However, with the described window partitioning, the boundaries between blocks in the output image would be clearly visible. Thus, in the general case, interpolation is necessary to avoid the occurrence of this effect. For this, the position of the currently considered pixel with respect to the four neighbouring window centres is determined (when possible) and bilinear interpolation is performed. An example of this situation is shown in Figure 7. The considered pixel has 4 neighbours in it: upper left (UL), upper right (UR), bottom left (BL) and bottom right (BR). By $s, t, w, z$ the distances of this point from its neighbours are specified. The new (output) pixel value is defined by Equation (6),

$$I_{new} = \frac{s}{s+w}\left(\frac{t}{z+t}f_{UL}(n) + \frac{z}{z+t}f_{BL}(n)\right) + \frac{w}{s+w}\left(\frac{t}{z+t}f_{UR}(n) + \frac{z}{z+t}f_{BR}(n)\right), \tag{6}$$

where $f$ denotes the normalised cumulative distribution function.

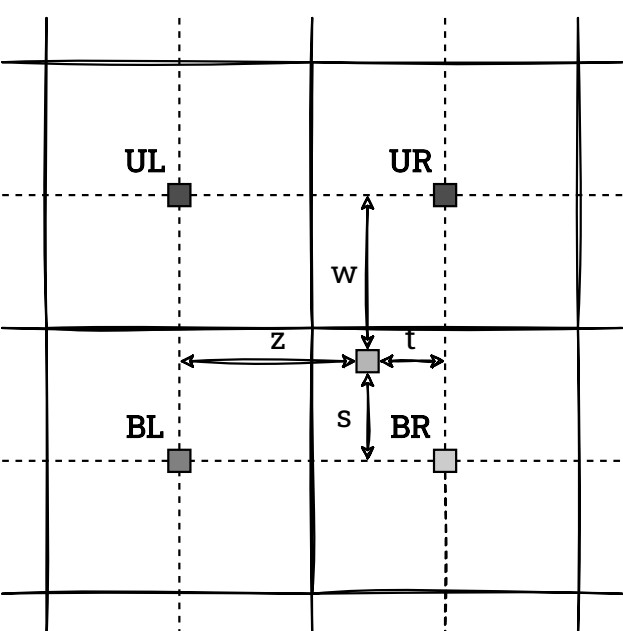

**Figure 7.** Bilinear interpolation scheme.

When a pixel is located on the edge or in the corner of an image (see Section 2.1 and Figure 3), it is impossible to determine 4 neighbours for a pixel. In the first case, only 2 neighbours can be selected. The mapping is then limited to linear interpolation. This situation is shown in Figure 8. The output pixel takes the value calculated according to Equation (7),

$$I_{new} = \frac{s}{s+w}f_{i,j-1}(n) + \frac{w}{s+w}f_{i,j}(n) \tag{7}$$

where $i, j$ denote the coordinates of the bottom pixel.

In the second case, i.e., when the pixel is located in the corner of the image, the only neighbour is the centre of the window with the analysed pixel. Then, the mapping is performed directly based on the mapping function, according to Equation (8),

$$I_{new} = f_{i,j}(n) \tag{8}$$

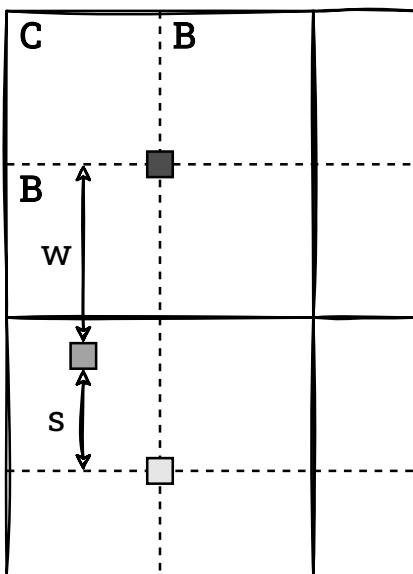

**Figure 8.** Linear interpolation scheme (for pixels on the image edge). C stands for corner region and B for border region.

### 2.5. Applications of CLAHE

As mentioned in Section 1, the CLAHE algorithm is frequently used for image contrast enhancement. In this subsection, we present selected applications published in the scientific literature.

The first group is medical image processing in a broad sense. In the paper [11], a two-step system named N-CLAHE for improving the quality of X-ray images was proposed. It used a global and a local approach (CLAHE). In the paper [12], a system for the detection of Covid-19 from lung radiograph images was presented. It used a deep convolutional neural network (VGG16) and the CLAHE method in pre-processing. Other medical applications of the CLAHE algorithm include image quality improvement systems for: fundus images [13,14], ultrasound images [15], and mammographic images [16].

The second large group of the CLAHE applications is the improvement of underwater images, which are characterised by low contrast and degraded shading quality. This approach was used in the works of [17–19]. Other applications of the CLAHE algorithm include the enhancement of thermal images [20], face recognition systems [21], systems to improve images captured in adverse weather conditions (fog, drizzle) [22], and improving image quality for digital side mirror cameras in vehicles [23].

Summarising this brief overview of the CLAHE applications, two aspects are worth noting. First, the method is used as a part of fully automatic systems as well as for the purpose of improving the quality of images analysed by a human—for example, a radiologist. Second, in at least some of the solutions, real-time operation with reduced energy consumption is desirable. Examples include systems for self-driving cars or underwater robots, but also medical equipment. Therefore, the work on the hardware implementation of the CLAHE algorithm in reprogrammable devices seems to be fully justified, and the results obtained can be applied to real-life solutions.

## 3. Related Work

The issue of accelerating computations for the CLAHE algorithm by using FPGAs is the content of many research papers. We present a selection of these below to provide context for our research. In Section 5, we also compare our results with those obtained in the described papers.

In the paper [10], the authors presented an implementation of the basic version of the CLAHE algorithm in an FPGA device. They treated the image as a whole, without dividing it into smaller blocks. In this way, they avoided interpolation, whose hardware implementation consumes additional resources. In this approach, a histogram was generated for each pixel based on its local neighbourhood. This was then used to construct a mapping function. However, the biggest drawback of this solution is the difficult data reuse scheme—the histogram must be updated for the "sliding window". The authors implemented this system in the Xilinx XC4VLX160 chip. It processed images with a resolution of $640 \times 480$ pixels in real-time. According to the authors, the proposed architecture is also capable of real-time processing of images with higher resolutions (even $1920 \times 1080$ pixels).

The problem of image partitioning into blocks and the associated interpolation is addressed in the work of [24]. The authors proposed some modifications to the redistribution process. In their framework, they moved a part of the computation to the histogram construction stage. They also introduced a look-ahead mechanism for redistribution, which makes it possible to reduce the number of iterations needed. Furthermore, they reformulated the interpolation step to reduce the use of hardware resources. They implemented the algorithm in an FPGA device for a $640 \times 480$ pixels video stream and achieved a processing speed of more than 354 fps. According to the authors, the proposed architecture can also process HD ($1280 \times 720$ pixels) and Full HD ($1920 \times 1080$ pixels) images at 90 fps and 33 fps, respectively.

A similar solution was presented by the authors of the paper [25]. Their system also tightly integrates the process of histogram generation and pixel redistribution. However, it is characterised by dividing the image into blocks that are not squares (size $64 \times 16$ pixels). This is performed to save memory resources on the chip. The authors did not provide a detailed description of the system architecture. In particular, there is no information how they combined redistribution with histogram generation and what interpolation precision they used. They implemented the proposed architecture in Xilinx's FPGA XC7Z045 FFG900-2 chip. For a video stream of $512 \times 512$ pixels, they achieved a system frequency of 129 MHz.

An implementation of the CLAHE algorithm for a high-resolution video stream was prepared by the authors of the paper [26]. In fact, they considered a modified version of this algorithm, referred to as adaptive histogram equalization with dynamic clip threshold (AHEwDC). The main difference is how the clip threshold was determined. Instead of a fixed value for all brightness levels, the authors varied them based on the features of the input image. Furthermore, they proposed a mean spatial filtering for the resulting mapping function (CDF). The changes made were intended to prevent the amplification of unwanted noise, especially in homogeneous image areas (e.g., sky). The authors did not provide details of the proposed system architecture in the part responsible for the implementation of the AHE. For evaluation purposes, they implemented it on an Intel Cyclone V FPGA device (5CSEMA5F31C6). The system achieved a maximum operating frequency of 75.48 MHz and was able to process a Full HD ($1920 \times 1080$ pixels) video stream at 30 fps.

A slightly different way of implementing the CLAHE algorithm on an FPGA chip was proposed by the authors of the paper [23]. They used a High-Level Synthesis (HLS) tool to implement the hardware architecture. They based it on a software implementation of the CLAHE algorithm available in the Open Computer Vision (OpenCV) library. It was necessary to introduce some modifications to it, e.g., by using appropriate HLS pragma instructions. The hardware architecture obtained was implemented on a PYNQ Z1 device. They considered two test cases for video streams with resolutions of $512 \times 512$ and

$1920 \times 1080$ pixels. In both cases, they obtained systems with performance competitive with implementation [24], which was done directly in Verilog HDL.

The works presented above demonstrate an interest in using FPGAs for the acceleration of the CLAHE algorithm. The authors proposed several different modifications to the algorithm to enable efficient hardware implementation. It is worth noting that the best architectures developed so far allow real-time processing of only Full HD ($1920 \times 1080$ pixels) video streams @ 30 fps. We are not aware of any work that has attempted a real-time implementation of the CLAHE algorithm for a 4K UHD ($3840 \times 2160$ pixels) video stream @ 60 fps.

## 4. Hardware Implementation

For the hardware implementation in this work, we use the ZCU 104 platform with AMD Xilinx Zynq UltraScale+ MPSoC device. To generate the video pass-through, we use Xilinx's example design (https://docs.xilinx.com/r/3.1-English/pg235-v-hdmi-tx-ss/Example-Design (accessed on 29 June 2022)).We transmit the input video stream from the computer via High-Definition Multimedia Interface (HDMI) 2.0 and decode it on the board. We perform all calculations only in the progammable logic part (PL, FPGA part of the heterogeneous device). The resulting image is also transmitted via HDMI 2.0 and displayed on the LCD screen. In Figure 9, we show a simplified schematic of our system that we use in this project. We develop and test all elements of the CLAHE algorithm by using Verilog HDL in Vivado Design Suite IDE 2020.2.

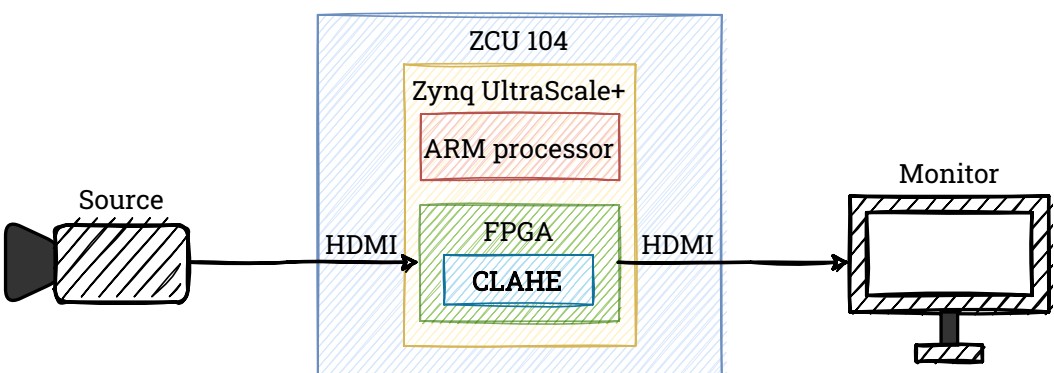

**Figure 9.** Simplified scheme of the system architecture. We use an HDMI source (camera/PC) and sink (UHD LCD monitor). All computations are performed in the PL/FPGA part of the used Zynq UltraScale+ MPSoC device.

Processing images in a well-known 1 pixel per clock format for a 4K video stream at 60 fps requires a clock frequency of about 600 MHz, which is close to the limit of modern FPGAs and can lead to routing problems or violating timing constraints. The solution to this problem is the use of a vector format—processing several pixels simultaneously. This format can be denoted as X ppc (X pixels per clock), which means processing X pixels in parallel in one clock cycle.

In our design we use 4 ppc format, which allows us to lower the clock frequency to about 150 MHz, at which point the problems described above do not occur. This frequency corresponds to a Full HD stream ($1920 \times 1080$ pixels) @ 60 fps in 1 ppc format. It should be noted, however, that our hardware modules can be easily adapted to other X ppc formats as well, as long as $\log_2(X) = c \in \mathbb{Z}$. With the data format used, 4 pixels appear simultaneously on the input, so 4 pixels must also be fed to the output in each clock cycle. This can cause various additional problems, depending on the type of operation implemented. In the case of context-free operations like colour space conversion, gamma correction etc., it is usually sufficient to duplicate the calculations performed. This leads to higher hardware resource consumption, but does not require additional logic.

The situation is slightly different in the case of context operations. Most often, they require not only the duplication of calculations, but also special mechanisms of data synchronisation, which increases the consumption of hardware resources even more [27]. Interesting and challenging cases are also to be found in algorithms with interdependences in the X ppc vector like Semi-Global Matching stereo computation or Connected Component Labelling [28]. In the described version of the CLAHE algorithm, there are no "typical" contextual operations; however, the calculation of histograms for several pixels at once requires some additional operations, which we describe in the following subsections. The block diagram of our implemented version of the CLAHE algorithm is shown in Figure 10.

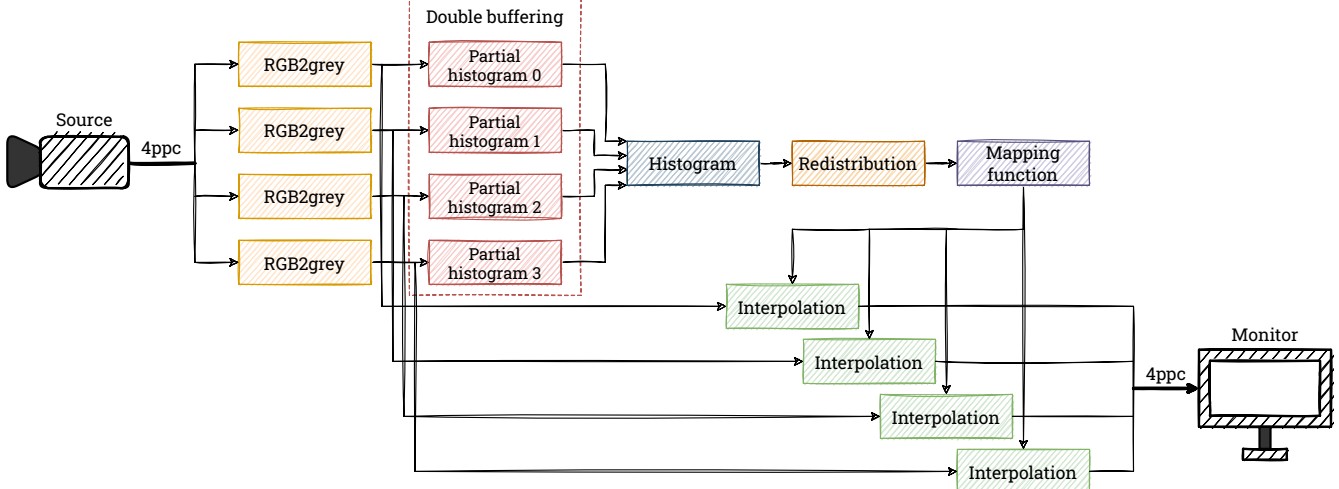

**Figure 10.** Block diagram of the implemented CLAHE algorithm in the 4 ppc mode. For a stream in the 4 ppc format, a conversion to greyscale (RGB2grey) is performed 4 times simultaneously and 4 partial histograms with 4 partial excesses (omitted from the diagram for clarity) are calculated. The histograms are then integrated, a redistribution is performed, and a mapping function is determined—these steps do not require working with a vector format. The final step—interpolation—is again performed on the 4 ppc stream.

As we mentioned earlier, in this project we focus on equalising the histogram for a single-component image. The input video stream uses the RGB format, so a conversion to greyscale is necessary. In each clock cycle, 4 RGB pixels are converted into 4 greyscale pixels, which are then fed into the histogram calculation module.

### 4.1. Generation of Tiles

As we described in Section 2.1, there are two main ways to generate blocks from the image. In our implementation, we use the approach that divides the image into non-overlapping rectangular windows, as it is in our opinion simpler in terms of hardware design and provides comparable results. For the behavioural simulation stage in Vivado, we use a test image of $256 \times 256$ pixels, which we divide into a grid of $4 \times 4$ blocks of $64 \times 64$ pixels. For the final hardware system, due to the larger input image size of $3840 \times 2160$ pixels, we decided to use a grid of $8 \times 8$ blocks of $480 \times 270$ pixels. We use simple pixel position counters in the image to determine the pixel's membership to a given block. Additional registers are used to store the coordinates of the block to which the analysed pixel belongs.

It should be mentioned that as input data arrives, the position counter (horizontal) is incremented by 4—due to the 4 ppc format used. In the general case, it may happen that pixels arriving in the same clock cycle belong to neighbouring blocks. However, in our solution, we choose such block dimensions (width divisible by 4) that the described situation does not occur. Additionally, in the resolution under consideration, a possible

offset by 4 pixels does not matter much. The 4 input pixels are then sent to the histogram generation module, together with the determined indexes of the block to which they belong.

When implementing the algorithm in 4K resolution, there is an additional issue to consider—the number of blocks into which the image is divided. A single block (for a single window) must contain memory for the mapping function (used during interpolation for the next frame). Assuming, as in the behavioural simulation, a window size of $64 \times 64$ pixels, we get $60 \times 34 = 2040$ blocks (and, therefore, 2040 blocks of memory). In our case, one such memory consumes 64 LUTs and 41 FFs, which for 2040 blocks equals 130,560 LUTs and 83,640 FFs. It is worth noting that these resources are consumed only by the memory for the mapping function—without histogram calculation, redistribution, interpolation, and video pass-through. With so many tiles, our CLAHE algorithm implementation would consume most of the available resources of the FPGA platform, making it unsuitable for being a component of a bigger vision system. Therefore, to reduce their consumption, we divided the image into $8 \times 8$ blocks, so each block had dimensions of $480 \times 270$ pixels.

*4.2. Histogram Calculation*

Due to the data format used (4 ppc), we generate 4 sub-histograms in each window, calculated separately for the input pixels, which are summed later in the value reading phase. Assuming a minimum division of the 4K resolution image into $8 \times 8$ blocks, the size of a single block is $480 \times 270$ pixels, or 129600 values. Because a partial histogram is generated, this number should be divided by the number of pixels per clock cycle (4). Then, the maximum value of a single bin is 32,400—its hardware representation can be stored on 15 bits (an unsigned integer). Moreover, the clip limit used in the CLAHE method further constrains the maximum value. The choice of the value of the parameter $b$ (see Equation (3)) is not obvious, so it is assumed that in the extreme case the method can be implemented like AHE (i.e., $b = 1$). Thus, 4 memories of 256 values $\times$ 15 bits are needed for each block.

Simultaneously with the histogram generation, we compute the excess. This is an improvement over the basic version of the algorithm (see Section 2), proposed in the paper [24]. Theoretically, all pixels can belong to the excess (which is obviously not a correct situation), but 15 bits are also allocated for the excess registers—there must be four, one for each pixel.

We use a dual-port Block RAM (BRAM) to implement the histogram calculation. This allows the calculation to have the lowest possible latency (one port is used for writing, the other for reading). The smallest memory unit in the FPGA device is 18 bits $\times$ 1024 words (18 Kb module). Due to the need to write four values simultaneously, 4 such memory units of $256 \times 15$ bits are needed for a single block. However, purely theoretically (without taking into account hardware limitations), a single 18 Kb module is sufficient for this purpose. Therefore, by using the BRAM in this way would be far from optimal.

The solution to this problem is to share BRAM memory for adjacent blocks containing pixels lying in the same image lines—for greater readability, we refer to these as a single window line. Then, depending on the number of windows, BRAM resources can be saved. Therefore, in the implementation, we use a common BRAM for the entire window line. In addition, we use the ping-pong buffering technique (double buffering)—one module is used to compute histograms from the current line of blocks, and the other module sends data from the previous line to subsequent modules at that time. Once the processing of the line of blocks is complete, the two modules swap roles. A diagram of the described double buffering is presented in Figure 11.

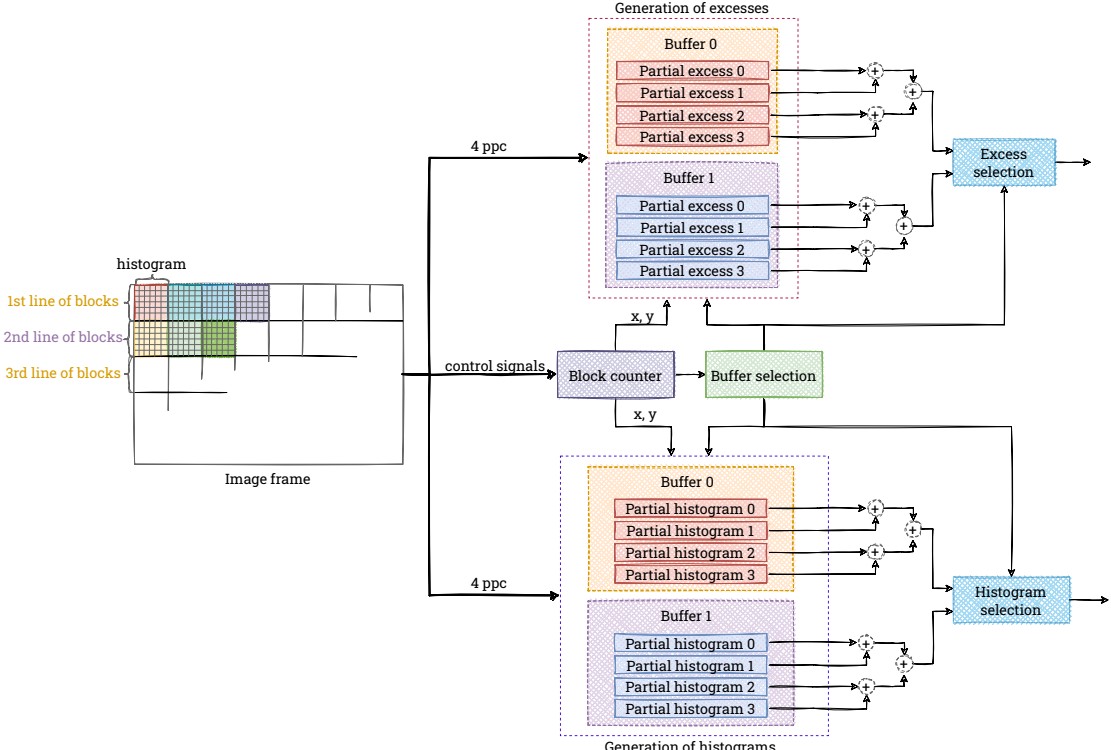

**Figure 11.** Block diagram of the implemented histograms' calculation. We use 4 partial histograms and excesses (one for each pixel from the 4 ppc), which are then aggregated by using summation trees. They are grouped into two buffers – one for the odd lines of the image and the other for the even lines. Thanks to that, we can simultaneously read the prepared data from one buffer and perform further calculations using the other.

The calculation of the histograms is solved in a "typical" way, including the handling of the occurrence of series of the same values. As already mentioned, due to the 4 ppc format, in the first phase, we calculate 4 partial histograms and at the same time 4 excess values. When reading from memory, we add them and create a final histogram, which is immediately processed in subsequent modules. It should be noted that the reading is combined with the resetting of the partial histograms (setting the values to 0), which prepares them to handle the next input data.

### 4.3. Redistribution

The input to the module is the histogram with the corresponding excess value. In the first step (first iteration over the histogram), we count the number of bins whose value is $\beta$. This parameter is used for the first iteration of the redistribution. This is another improvement of the basic version of the algorithm (see Section 2), proposed in the paper [24]. The value of the parameter $m$ is then not $excess/N$, but $excess/(N-NFB)$, where: $N$ is the number of all bins of the histogram, and $NFB$ is the number of bins that are filled (at the limit). This increases the excess value, which is redistributed during the first iteration.

When computing the parameter $m$, we use another improvement proposed in the paper [24]. We store the remainder from dividing $excess/(N-NFB)$ as the value $e$. It represents the part of the excess that would remain to be redistributed after adding $m$ to each "free" bin, according to the basic version of the algorithm (see Section 2 and Listing 1). However, we reduce it already in the first iteration. Then, to each "free" bin we add the value $m+1$ (1 from the remainder $e$, which is somehow a combination of two redistribution steps). If the result exceeds $e$, then we only add $m$ or a value smaller than $m$, until the set limit is filled. In the last mentioned situation, we also modify the parameter $e$, adding the

part of *m* that did not fit into the histogram bin. The histogram obtained after this step is written to a temporary buffer in Distributed RAM (DRAM).

To implement this iteration, we use a state machine that handles the described steps. After receiving the information that the histogram generation has finished (the signal sent from the previous module), we compute $NFB$ and simultaneously buffer the histogram values. In the second step, we compute the values of *m* and *e*—this requires division, which we implement by using the IP Core from AMD Xilinx. In the third step, we perform the first iteration of the redistribution by using the calculated values.

Subsequent iterations of the redistribution are simpler. In them we increment the "free" bins of the histogram only by 1. This is accompanied by an analogous decrease of the *e* value transferred from the previous iteration. Technically, this is done by reading from one histogram buffer, correcting the value (if possible) and writing to the next buffer. In this way we realise additional three iterations of the redistribution. This is sufficient to distribute the excess in most typical cases.

We implement each of these as a simple state machine, similar to the first iteration. In this case, it has two states. In the first one, we wait for the previous iteration to finish (the high state of the relevant signal). In the second one, we perform the actual redistribution process as described.

We present all the described operations in Figure 12. It is worth noting that we implement the redistribution process in a universal way. Without much effort, it can be easily modified to perform slightly differently, as required. For example, additional iterations can be applied, as well as an adaptive clip limit.

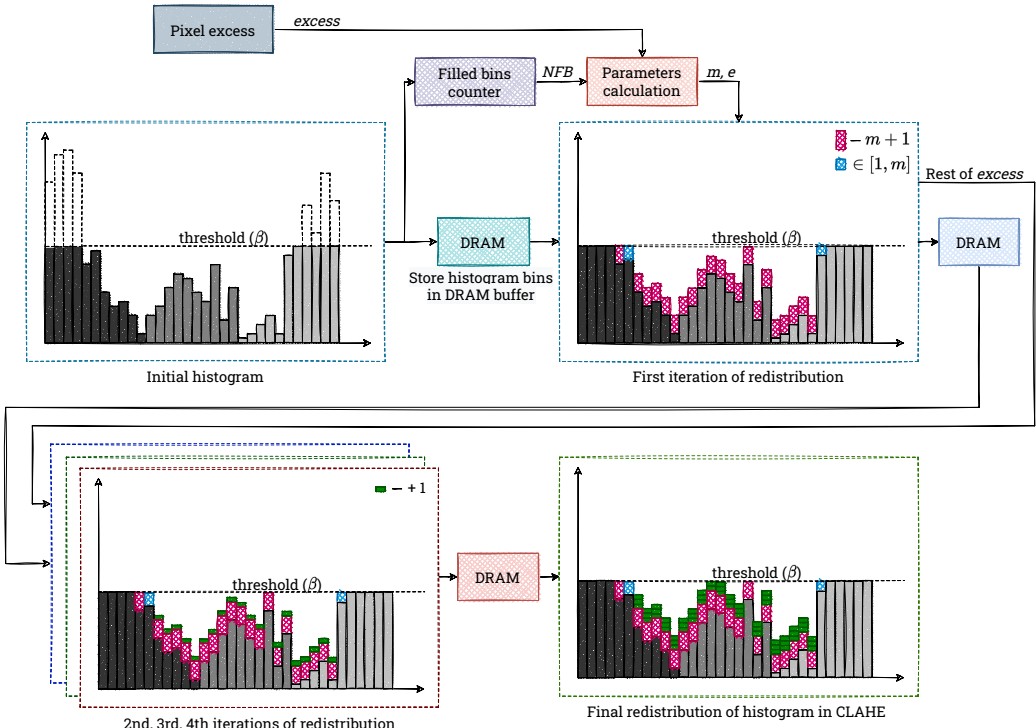

**Figure 12.** Scheme of the implementation of the redistribution process. We read the input histogram sequentially (bin after bin), count the bins filled to the limit *β*, and combine this value together with the input excess to calculate *m* (average excess) and *e* (residual). The input histogram is then stored in the DRAM buffer. After calculating the mentioned parameters, we read the histogram from the DRAM buffer and perform the first iteration of the redistribution, in which we try to add $m + 1$ to each bin. If it is not possible, we fill the bin to the limit *β* and accordingly modify the value of *e*. The resulting histogram is stored in another DRAM buffer. After that, we perform the next 3 iterations, this time adding only 1 pixel to each bin (thus decreasing the value of *e*), where it is possible. Between each iteration there are also DRAM buffers (not present in the diagram for clarity). In this way, we finally obtain a redistributed histogram.

### 4.4. LUT-Based Mapping Function

After the last redistribution iteration, we determine the mapping function for a given window. First, we calculate the cumulative histogram. Then, we normalise each value by dividing it by the number of pixels in the window, and multiplying by 255 (the upper range of pixel values represented on 8 bits). The results are stored in four buffers implemented in DRAM. Their number results from the use of the 4 ppc format, which requires a simultaneous access to 4 pixels in the following interpolation. We use the coordinates of the block to which the histogram belongs as its address and duplicate them to 4 buffers. In this way, the values for the 4 considered pixels are easily available for interpolation.

### 4.5. Interpolation Method

The final element of the implemented algorithm is a bilinear interpolation. This module operates on the next frame from the video sequence—the histograms are calculated on frame $N$, while the interpolation and calculation of output pixel values are executed for frame $N + 1$. This approach is justified by the real-time operation of the algorithm (60 frames per second), so that the differences between adjacent frames are small. Performing computations for the current frame is also possible, but it requires additional buffering of the frame, partial or total (depending on the concept adopted), which complicates the system and increases power consumption with practically no improvement in image quality.

To perform bilinear interpolation, it is first necessary to determine to which block the input pixels belong to. It is realised by simple position counters in the image. They are also used to determine the neighbours of the pixels under consideration (4, 2, or 1). We read the values corresponding to each of the 4 pixels (for the 4 ppc format) from the LUTs of the selected blocks and pass them to the input of the interpolation module—each pixel to a separate one. For pixels in the corner or on the edge of the image, only 1 or 2 values are passed to the interpolation module, depending on the position in the image, as described in Section 2.4.

Next, we calculate the values $s, t, w, z$, and then, according to Equation (6), the pixel value using a set of adders and multipliers operating in parallel. Next, we divide it by the dimensions of the window ($XX \cdot YY$, which for pixels with 4 neighbours corresponds to $(s + w) \cdot (z + t)$) to map it to the brightness interval 0–255. It is worth mentioning that for pixels on the edge or in the corner of the image, the output value is also divided by the window dimensions. Finally, we pass the calculated value of each of the 4 pixels to the output of the interpolation module, after which we combine them into an output vector and display on the monitor as the result of the CLAHE algorithm.

## 5. Results

In the algorithm testing stage, the effect of different parameters on the performance of the CLAHE algorithm was checked on selected test images. One of them is presented in Figure 13 with the results of the algorithm for different divisions of the input image into windows—starting from 4 × 4, through 8 × 8, ending with 16 × 16.

As already mentioned in Section 4, the CLAHE algorithm was implemented on the ZCU 104 development board with AMD Xilinx Zynq UltraScale+ XCZU7EV-2FFVC1156 MPSoC device. A comparison of the most important features of our solution with other works that also implemented the CLAHE algorithm on FPGA platforms is provided in Table 1. It also compares the hardware resource usage of the CLAHE module (without additional components) with other works. Our solution features by far the highest resolution of the processed video stream.

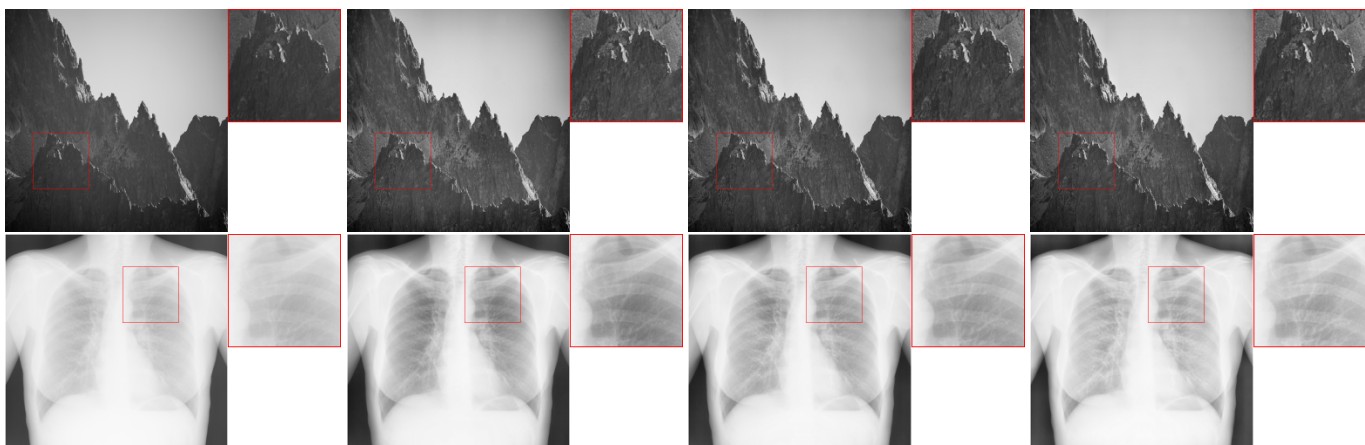

**Figure 13.** Example results for test images. The figures show successively (from the left) the original image, the result of the CLAHE algorithm for 4 × 4 blocks, the result of the CLAHE algorithm for 8 × 8 blocks, and the result of the CLAHE algorithm for 16 × 16 blocks.

Typically, for an FPGA implementation of the algorithm, some hardware resources are allocated for the video pass-through needed to correctly receive and send the video stream. The resource consumption for the CLAHE module only and the complete algorithm (with the video pass-through), processing a real-time 4K video stream, is provided in Table 2.

FPGA platforms enable efficient real-time processing of large amounts of data with low power consumption. According to estimation in the Vivado Design Suite IDE, the CLAHE algorithm implemented on the ZCU 104 platform for 4K resolution needs only 4.83 W of power. The entire system is capable of operating at a maximum clock frequency of 151 MHz (value is estimated with Vivado IDE); therefore, our system works at nearly maximum possible frequency (150 MHz). The photo of the proposed system that allows real-time adaptive image histogram equalisation by using the CLAHE method in 4K resolution is shown in Figure 14.

**Table 1.** Comparison of the most important parameters of hardware implementations of the CLAHE algorithm on FPGA platforms and resource utilisation for the CLAHE module. Only our solution supports the 4K resolution, but despite processing 4 pixels at once, the resource utilisation is comparable to Full HD solutions, e.g., [23]. Low memory utilisation of our solution is also worth noting, which was possible to achieve by using the ping-pong buffering technique during histogram calculation. The utilisation of DSP resources in not presented, as it was rarely reported in other works.

| Implementation | Platform | Resolution | FPS | Frequency [MHz] | # of LUTs | # of Flip-Flops | # of Block RAMs |
|---|---|---|---|---|---|---|---|
| Kokufuta [10] | AMD Xilinx XC4VLX160 | 640 × 480 | 538 | 209 | 43,915 | - | 192 |
| ine Unal [24] | AMD Xilinx Zynq-7000 | 640 × 480 | 354 | 109 | 4766 | 440 | 16 |
| ine Unal [24] | AMD Xilinx Zynq-7000 | 1920 × 1080 | 33 | 69 | - | - | - |
| ine Kim [25] | AMD Xilinx XC7Z045 | 512 × 512 | 492 | 129 | 98,945 | 85,600 | 8 |
| ine Xu [26] | Altera Cyclone V | 1920 × 1080 | 30 | 76 | 14,807 * | 4794 | 9 |
| ine Honda [23] | AMD Xilinx PYNQ Z1 | 1920 × 1080 | 47 | 111 | 29,800 | 38,500 | 33 |
| ine **This work** | **AMD Xilinx ZCU 104** | **3840 × 2160** | **60** | **150** | **30,972** | **21,178** | **16** |

\* This value cannot be directly compared to the rest because the Logic element in Intel's chips is different from Xilinx's one.

**Table 2.** The use of hardware resources for the CLAHE algorithm on the ZCU 104 board. Note a relatively low resource utilisation of our module. It consumes even less LUTs and Flip-Flops than basic video pass-through, which is needed to provide the input and output images. Therefore, our implementation can be used in combination with other hardware modules of the entire embedded vision system as it consumes only a small part of the available resources. The utilisation of DSP resources is not reported as our implementation of the CLAHE module uses none of them.

| Resource Type | Available | Pass-Through | CLAHE Module | Full Algorithm |
|---|---|---|---|---|
| LUT | 230,400 | 38,097 (17%) | 30,972 (13%) | 68,932 (30%) |
| ine Flip-Flop | 460,800 | 44,673 (10%) | 21,178 (5%) | 63,703 (14%) |
| ine Block RAM | 312 | 7 (2%) | 16 (5%) | 23 (7%) |

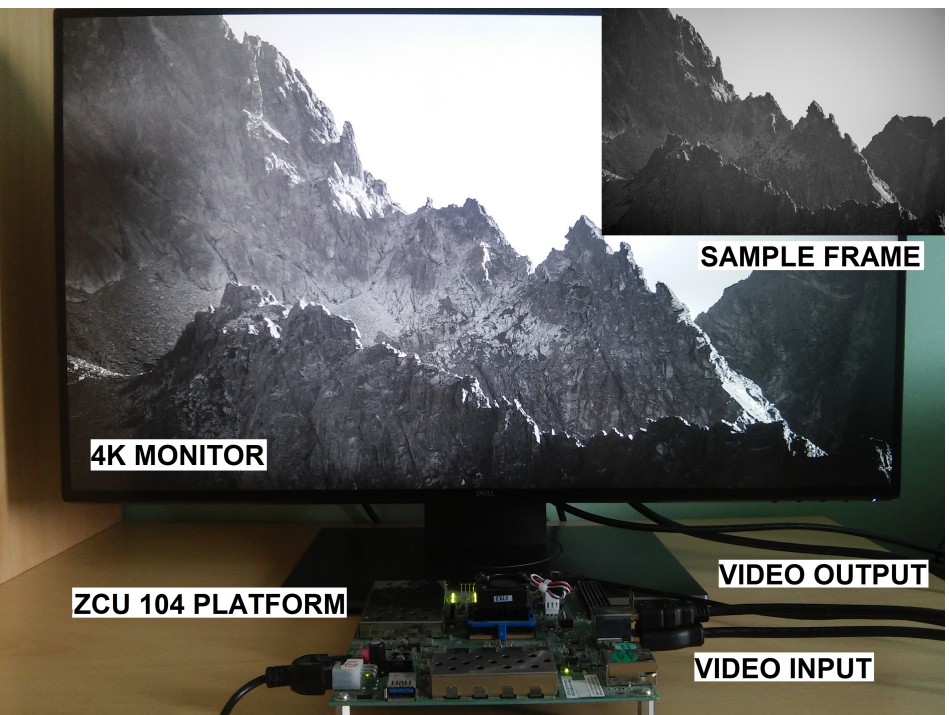

**Figure 14.** Photo of the proposed system in operation. The input video signal is transmitted from the source (a computer) to the ZCU 104 board, equipped with the AMD Xilinx Zynq UltraScale+ MPSoC chip. The output image—after applying the CLAHE algorithm—is transmitted and displayed on a 4K monitor.

## 6. Discussion

In our implementation, we use a slightly modified version of the "classical" CLAHE algorithm (see Section 2). The changes we have made (in terms of the algorithm, not the hardware architecture) mainly involve the redistribution process. Following the suggestions from the paper [24], we shortened the redistribution process by determining the parameter $m$ based only on incomplete histogram bins and introduced the addition of the value $m + 1$ during the first iteration ("future lookup"). It is worth noting, however, that other proposals for modifications of the CLAHE algorithm can also be found in the literature.

The first of these also applies to the redistribution process and comes from the OpenCV library [29]. In this implementation of the CLAHE algorithm, the preset $\beta$ limit is only respected during the histogram generation phase. During the redistribution, the values of particular bins are incremented independently of their current state. The result of such an operation is shown in Figure 15.

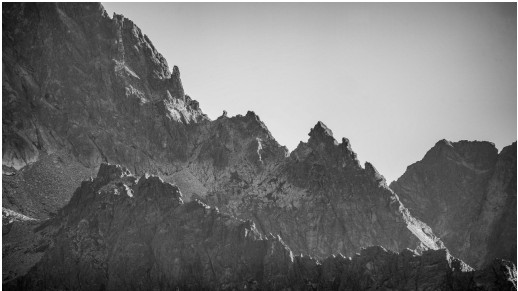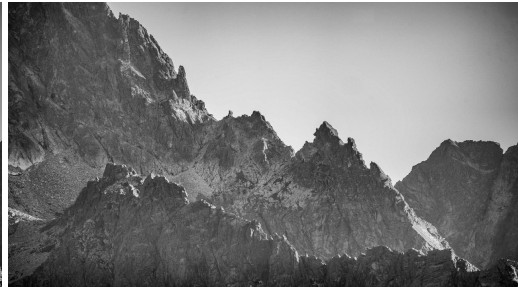

**Figure 15.** Comparison of output images obtained with the "classic" (**left**) and OpenCV (**right**) redistribution. In this case, the redistribution from OpenCV results in a brighter image.

The implementation of this solution in an FPGA for a 4 ppc stream is not a major problem, especially using the proposed redistribution module. Moreover, the OpenCV variant can be considered more hardware-friendly than the "classical" approach. First, in this case, only two redistribution phases are sufficient. In addition, the lack of a $\beta$ limit simplifies the control logic, which enables further reduction in hardware resource consumption. Despite the advantages mentioned above, we chose not to implement this solution because it clearly deviates from the assumptions of the "classic" CLAHE algorithm. In case of hardware implementation, this approach may cause additional problems as in FPGAs the values are usually represented by a fixed number of bits. The permission to exceed the limit may lead to incorrect values (when the limit is exceeded) or the need to increase number of bits used for data representation, thus using additional hardware resources.

The second possible modification of the algorithm concerns the output image preparation step. It requires combining the data generated in tiles. The simplest solution is to treat them as separate image fragments. However, this leads to visible artefacts on the borders of particular tiles. For this reason, bilinear interpolation is commonly used (in our work as well). However, it is quite computationally complex. Some alternative may be to average the values of the cumulative distribution function in the context of $3 \times 3$ blocks, as proposed in the paper [26].

A comparison of the results obtained with all three methods is presented in Figure 16. Bilinear interpolation gives the best qualitative results. In this case, the boundaries between blocks are virtually invisible. As expected, they are most visible in the absence of the interpolation. The use of the average of the cumulative distribution function can be considered as an intermediate solution. Depending on the tile size used, these artefacts are more or less visible.

In the case of a hardware implementation, the simplest approach is not to perform the interpolation. Then, no additional control logic is needed and the output image generation is reduced to the use of basic lookup table elements. However, the results obtained with this approach are not very satisfactory. For this reason, in this paper we have used bilinear interpolation, which compensates best for the differences between the blocks. It is worth noting that our implementation is relatively easy to convert into a CDF averaging module. As with bilinear interpolation, it requires access to several elements of the lookup tables simultaneously. The main difference, however, is the way of data processing after reading from the memory—it is simpler and less computationally intensive than in the module we implemented.

The third possible modification of the algorithm is to adaptively select the $\beta$ limit, depending on the parameters of the tile. One method of this type has been proposed in the paper [30]. There, the $\beta$ limit is computed according to Equation (9),

$$\beta = \frac{M}{N}\left(1 + P\frac{l_{max}}{R} + \frac{\alpha}{100}\left(\frac{\sigma}{Avg + c}\right)\right) \tag{9}$$

where: $M$ is the number of pixels in the window, $N$ is the number of histogram bins (typically 256), $l_{max}$ is the maximum value of the brightness level in the tile, $R$ is the

maximum possible value of the brightness level (typically 255), $\sigma$ is the standard deviation of the brightness in a tile, *Avg* is the average brightness in the tile, *c* is the very small constant (as a safeguard against dividing by 0), and *P* and $\alpha$ are the coefficients controlling the weights of the components of the sum.

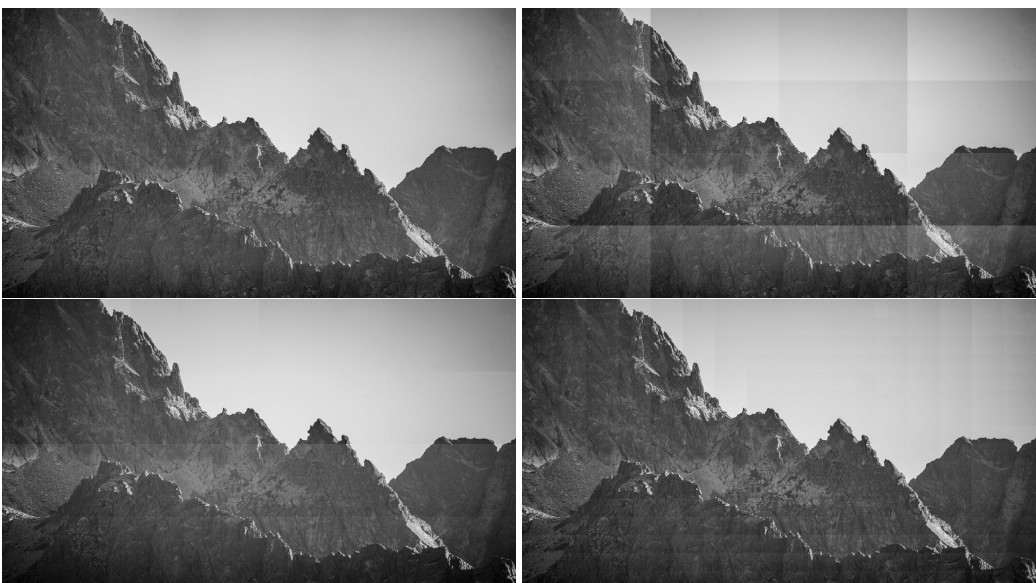

**Figure 16.** Comparison of different methods of generating the output image: bilinear interpolation (**top left**), no interpolation with $4 \times 4$ grid of blocks (**top right**), mean CDF with $4 \times 4$ grid of blocks (**bottom left**) and mean CDF with $16 \times 16$ grid of blocks (**bottom right**). Bilinear interpolation provides the best results with almost no visible artefacts.

The result of using the described adaptive limit $\beta$ is shown in Figure 17. The differences between this method and the "classical" solution are practically imperceptible for this image. At the same time, the hardware implementation of the adaptive $\beta$ limit, especially for a 4 ppc stream, is definitely more complicated. The biggest problem in this case is the calculation of the standard deviation. The reason for this is the requirement for the prior determination of the mean, which is only possible after reading all the pixels from a particular tile. Nevertheless, by using a technique similar to the generation of histograms, it would be possible to calculate partial averages and, from these, the final average. Then, by using this value and the created histogram, the standard deviation can be determined. In doing so, a single line of blocks needs to be cached (delayed) accordingly, which requires the use of a certain amount of BRAM memory (the size of the tiles is relatively large). Finally, determining the $\beta$ limit is quite straightforward, although it requires division and multiplication.

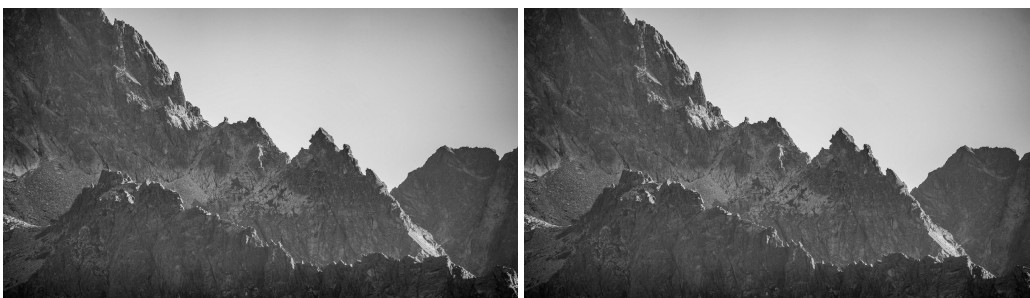

**Figure 17.** Comparison of output images obtained with constant (**left**) and adaptive (**right**) limit $\beta$. The difference between them is hardly noticeable for the considered image.

The next modification is not directly related to the CLAHE algorithm. It involves improving the quality of the input image by sharpening it with a Laplace filter, as mentioned by the authors of the paper [31]. An example result of this approach is shown in Figure 18. The output image has a slightly lower average brightness, which can be subjectively considered as a certain advantage.

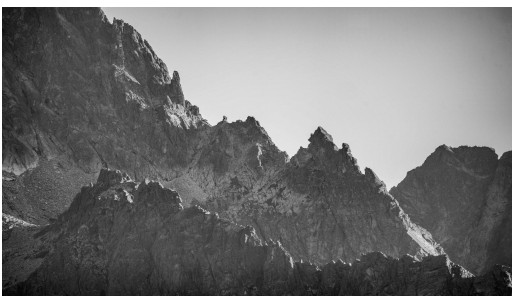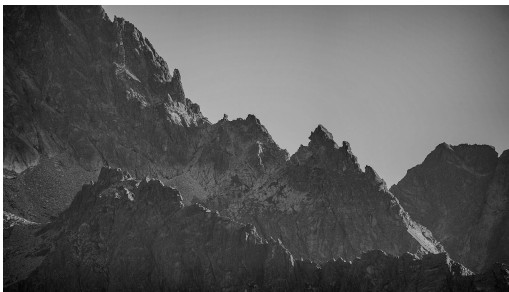

**Figure 18.** Comparison of output images obtained without (**left**) and with (**right**) a Laplace filter on the input image.

However, it may turn out that for some specific classes of images, the Laplace filter significantly improves the results of the CLAHE algorithm. In that case, the hardware implementation of such a solution reduces to preparing a separate module responsible for performing the filtering of the input image. The Laplace filter is a simple contextual operation. In view of this, its hardware implementation should not pose a problem [27].

The last thing we want to discuss concerns the use of the CLAHE algorithm for colour images. In this case, a couple of different approaches can be found in the scientific literature. These assume the equalisation of both selected components of the colour spaces and all of them at the same time. Examples of this type of approach include the use of the CLAHE algorithm on the V component of the Hue, Saturation, Value (HSV) model and on all components of the RGB model (either one selected or all at the same time). The results obtained in this way are presented in Figure 19. As can be seen, in some cases (e.g., all RGB components simultaneously), the CLAHE algorithm significantly contributes to improving the quality of the input image.

The hardware implementation of such a solution is not a problem, especially when using the prepared modules. Depending on the chosen approach, it requires additional conversions between colour spaces (rather easy to implement) and a possible multiplication of the CLAHE algorithm to process individual channels of the video stream.

In the discussion above, we have considered our hardware implementation in relation to possible modifications of the CLAHE algorithm that have been proposed in the scientific literature. As can be seen, the variant proposed in this work performs relatively well. At the same time, it provides a convenient basis to implement other variants of the CLAHE algorithm. Any modifications can be easily added by slightly changing the developed hardware modules or adding new ones (e.g., the Laplace filter or colour space conversions).

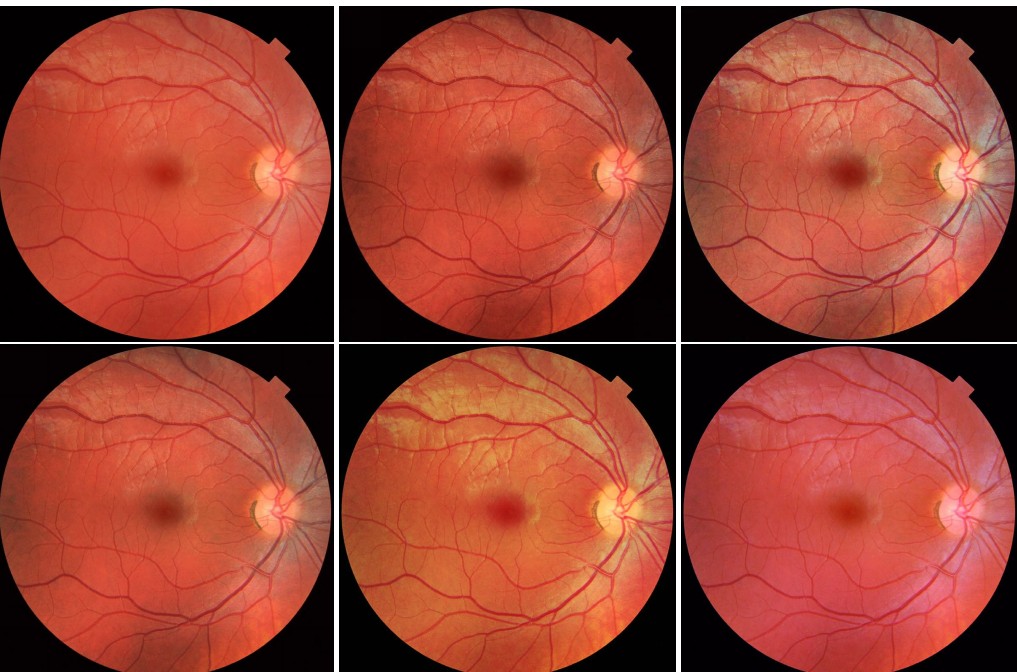

**Figure 19.** Results of the CLAHE algorithm for an exemplary colour image (**top left**). We use CLAHE on different image channels: V from HSV model (**top centre**), R, B and G from RGB model together (**top right**), and separately R (**bottom left**), G (**bottom centre**), B (**bottom right**).

## 7. Conclusions

In this work, we propose a hardware implementation of the CLAHE algorithm for a 4K UHD video stream (3840 × 2160 pixels @ 60 fps, 150 MHz clock). By parallelising the calculations on the SoC FPGA hardware platform used, we are able to achieve real-time processing (60 fps). To limit very high clock frequency of the FPGA device, we use the 4 ppc vector data format, with 4 pixels processed simultaneously. This generated the need to implement additional logic to calculate and sum the partial histograms for a given window. In the redistribution process, the maximum number of iterations is limited to 4 to allow this step to operate deterministically on the FPGA platform. We also use bilinear interpolation in the algorithm to eliminate visible boundaries between the blocks into which the input image was divided during histogram computation and redistribution.

In future work, additional improvements and modifications to the hardware version of the algorithm are possible, e.g., by integrating the histogram calculation and excess redistribution steps. It is also possible—in a very easy way—to extend the algorithm to colour images, by using the realised module of the CLAHE algorithm to all or selected components of the image (discussed in Section 6). One of the potential options is to prepare a "dynamic" version, in which the dimensions of the windows into which the image is divided, as well as the value of the $\beta$ parameter, can be changed during operation. The latter can also be modified in such a way that its value is set automatically during the algorithm operation. Moreover, our module can be integrated as a part of a bigger computer vision system, e.g., for traffic sign or pedestrian detection.

Potential applications of the realised algorithm include broadly the pre-processing of low-contrast images. Such methods are useful in autonomous vehicles and drones, especially in more difficult weather conditions. Popular applications also include thermal imaging or underwater photos. Medical images, such as X-rays or CT scans, are also an important application category, where contrast enhancement makes it easier for doctors to analyse the data and make an accurate diagnosis.

**Author Contributions:** The authors made the following contributions to this work: conceptualisation, T.K., K.B., H.S., and M.W.; methodology, T.K.; software, T.K., K.B., H.S., and M.W.; validation, T.K., K.B., H.S., and M.W.; formal analysis, T.K., K.B., H.S., and M.W.; investigation, T.K., K.B., H.S., and M.W.; resources, T.K., K.B., H.S., and M.W.; data curation, T.K, K.B., H.S., and M.W.; writing—original draft preparation, T.K., K.B., H.S., and M.W.; writing—review and editing, T.K., K.B., H.S., and M.W.; visualisation, T.K., K.B., H.S., and M.W.; supervision, T.K.; project administration, T.K.; funding acquisition, T.K. All authors have read and agreed to the published version of the manuscript.

**Funding:** The work presented in this paper was supported by the National Science Centre project no. 2016/23/D/ST6/01389 entitled "The development of computing resources organisation in latest generation of heterogeneous reconfigurable devices enabling real-time processing of UHD/4K video stream".

**Institutional Review Board Statement:** Not applicable.

**Informed Consent Statement:** Not applicable.

**Data Availability Statement:** Not applicable.

**Acknowledgments:** The authors would like to thank Aleksander Orlikowski and Krzysztof Grzyb for conducting initial research on FPGA implementation of the CLAHE algorithm.

**Conflicts of Interest:** The authors declare no conflict of interest.

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
