# Peer review of "Real-Time CLAHE Algorithm Implementation in SoC FPGA Device for 4K UHD Video Stream"

_electronics, doi:10.3390/electronics11142248_

Round 1
Reviewer 1 Report
This paper describes an implementation of CLAHE method in SoC FPGA. By using the Xppc format (4ppc), the authors realize 3840x2160 pixels with 60FPS. It is affordable to 4K UHD Video stream and the impact to the practical usage is large. Although no surprising techniques are used, the quality of implementation is high and it can be a good implementation example for engineers who try this field.
Author Response
Dear Reviewer,
We are very grateful for the time you have put into this review. We thank you for your nice comment and appreciation of our work.
Tomasz Kryjak
Assistant professor
AGH University of Science and Technology in Kraków
Reviewer 2 Report
In the paper “Real-time CLAHE Algorithm Implementation in SoC FPGA
Device for 4K UHD Video Stream" the authors made an honorable effort to implement the CLAHE algorithm in SOC FPGA for 4k UHD video stream.
- The scientific contribution is not in this version of the article very clearly presented.
- The authors' performance is primarily industrial, technical and less scientific.
- The presentation of the images with and without the application of the innovative elements would prove the usefulness of the proposed method.
- The work must be reorganized and Figure 9 together with Tael 1, 2 must not interfere with Section 6.
- I recommend that in each image to be suggested a region of interest (ROI) where an unsuspecting reader can see the benefit brought by algorithm.
- The references are very technical and do not offer a broader picture of the field which is also very rich.
- A Verilog module would be more useful in Listing 1 and possibly moved to an Appendix.
The paper must have academic accents and it cannot be reduced to a technical note related to an implementation of an algorithm. The fact that we are resorting to a recent technology is a technical skill and an approach with scientific accents is much more useful to the readers of the Electronics journal. I recommend major revision.
Author Response
Dear Reviewer,
We are very grateful for the time you have put into this review. We appreciate your comments and did our best to improve the paper. Most of the issues are addressed in the text and the changes are highlighted in blue colour. We also did some minor improvements to the language used in the paper. Please find below the responses to your particular remarks.
In the paper “Real-time CLAHE Algorithm Implementation in SoC FPGA Device for 4K UHD Video Stream" the authors made an honorable effort to implement the CLAHE algorithm in SOC FPGA for 4k UHD video stream.
Thank you very much for the appreciation of our work.
R2.1 The scientific contribution is not in this version of the article very clearly presented.
In our view, the main novelty is the hardware architecture that enables real-time processing of 4K/UHD video stream (@ 60 fps). This problem has not been addressed so far in the scientific literature. We have discussed this issue more extensively in our response to comment R2.2. In addition, following the suggestion in R2.8, we have added some scientific accents (see response to comment R2.8 for details). We also highlighted our contribution in Section 1 - Introduction.
Was:
The main contributions of this paper are as follows:
- a hardware implementation of the CLAHE algorithm on an FPGA platform, enabling real-time processing of a 4K (Ultra HD) video stream, which to our best knowledge has not been done before,
- using a vector stream format (4 ppc) to implement the CLAHE algorithm.
Is:
The main contributions of this paper are as follows:
- a hardware implementation of the CLAHE algorithm on an FPGA platform, enabling real-time processing of a 4K (Ultra HD) video stream, which to our best knowledge has not been done before,
- using a vector stream format (4 ppc) to implement the CLAHE algorithm, which should be considered as an architectural novelty due to required redesign of its components.
R2.2 The authors' performance is primarily industrial, technical and less scientific.
We agree that in the domain of hardware acceleration of algorithms the scientific contribution is in some cases “less dominant”. Our research should be considered as a follow-up of the previously papers on implementing CLAHE in FPGA [3] (FPL) [4] (FPL) [6] (ICEIC) [7] (IEEE Transactions on Circuits and Systems) [8] (ICEIC), which as indicated were published in conference proceedings and a journal. Our main contribution is the support for UHD/4K resolution. It was possible only when using a vector format - in our case 4 pixels per clock cycle. However, this involved the redesign of all modules of the CLAHE algorithm, especially histogram computation and image interpolation.
We have also added two new schemes, which present our modules:
- Figure 11 - Histogram calculation
- Figure 12 - Histogram redistribution
Summing up. Our contribution is similar to others in the FPGA community and in this particular case could be considered as architectural.
R2.3 The presentation of the images with and without the application of the innovative elements would prove the usefulness of the proposed method.
To address the comment, we have prepared an additional figure showing step-by-step the operation of the CLAHE method (Figure 2 in Section 2). In addition, we have supplemented the article with the description of several CLAHE variants (see response to comment R2.8 for details). However, we would like to point out that algorithmically there are no innovative elements - we tried to obtain a solution close to the version available in the OpenCV library.
R2.4 The work must be reorganized and Figure 9 together with Tael 1, 2 must not interfere with Section 6.
We added some additional figures and subsections, therefore, the entire paper was reorganised. Also, the paper with the changes highlighted in blue colour (as it is now) is different from the “clean” one. If the paper gets accepted for publication, the final version will be reorganised to avoid interference of tables and figures with particular sections.
R2.5 I recommend that in each image to be suggested a region of interest (ROI) where an unsuspecting reader can see the benefit brought by algorithm.
In Figures 1 and 13 we marked ROIs in the images and zoomed them to better present the differences caused by applying the CLAHE algorithm. In some cases they are easily visible, in others less.
R2.6 The references are very technical and do not offer a broader picture of the field which is also very rich.
We agree that in the first version of the article the references were very technical and limited only to previous implementations of CLAHE and two of our earlier articles that illustrated the problem of data analysis in the 4 ppc format. As suggested here, we have significantly enriched the literature.
Firstly, we added 6 articles on various FPGA applications in vision systems (as a broader context for the technology), which we discussed in Section 1 - Introduction:
FPGAs have found applications in many real-time vision systems. They are used for optical flow determination, e.g. with the Lucas-Kanade and Horn-Schunck methods [23], or stereo correspondence with the Semi-Global Matching algorithm [24]. They also enable the implementation of advanced object tracking methods [26]. FPGAs can be applied in advanced driver assistance systems, for example for high-speed gaze detection [27] and unmanned aerial vehicles, for example the Simultaneous Localization and Mapping [28]. In the last few years, a very large number of scientific and industrial works also addresses the topic of deep neural networks acceleration (especially convolutional neural networks) on FPGAs [29].
Secondly, we have added 12 articles on various applications of the CLAHE algorithm - in medical, underwater and other image processing. We have discussed these in a new subsection 2.5 - Applications of CLAHE (content below). In addition, we have added a reference to the very popular Digital Image Processing book by Gonzalez and Woods. We hope that this is the representation of the 'broader picture' that this comment was about.
As mentioned in Section 1, the CLAHE algorithm is frequently used for image contrast enhancement. In this subsection, we present selected applications published in the scientific literature.
The first group is medical image processing in a broad sense. In the paper [10], a two-step system named N-CLAHE for improving the quality of X-ray images was proposed. It used a global and a local approach (CLAHE). In the paper [12], a system for the detection of Covid-19 disease from lung radiograph images was presented. It used a deep convolutional neural network (VGG16) and the CLAHE method in pre-processing. Other medical applications of the CLAHE algorithm include image quality improvement systems for: fundus images [11], [15], ultrasound images [13] and mammographic images [14].
The second large group of the CLAHE applications is the improvement of underwater images, which are characterised by low contrast and degraded shading quality. This approach was used in the works of [17], [16] and [18]. Other applications of the CLAHE algorithm include the enhancement of thermal images [19], face recognition systems [20], systems to improve images captured in adverse weather conditions (fog, drizzle) [21], and improving image quality for digital side mirror cameras in vehicles [8].
Summarising this brief overview of the CLAHE applications, two aspects are worth noting. First, the method is used as a part of fully automatic systems as well as for the purpose of improving the quality of images analysed by a human - for example, a radiologist. Second, in at least some of the solutions, real-time operation with reduced energy consumption is desirable. Examples include systems for self-driving cars or underwater robots, but also medical equipment. Therefore, the work on the hardware implementation of the CLAHE algorithm in reprogrammable devices seems to be fully justified, and the results obtained can be applied to real-life solutions.
R2.7 A Verilog module would be more useful in Listing 1 and possibly moved to an Appendix.
The Verilog module that corresponds to Listing 1 is quite complex. We have attached it as a response to the review (supplementary file). In our opinion, including it in an appendix will not be particularly useful for potential readers, because while there is a lot of code, the implementation concept itself is rather simple and based on a few state machines.
Furthermore, in the FPGA community the practice of sharing HDL code (modules described in Verilog, SystemVerilog, or VHDL) is not very common. In our case, a limitation is the not-quite-precise interpretation of intellectual property protection - a description in HDL may be considered as a description of a hardware architecture (e.g. for an ASIC). In addition, years of experience have shown that the portability of designs created in the Vivado tool is usually very limited (in this case we would expect problems with proper BRAM configuration). Here we would need to provide the whole project, for a particular Vivado version and also a particular OS version (Ubuntu).
R2.8 The paper must have academic accents and it cannot be reduced to a technical note related to an implementation of an algorithm. The fact that we are resorting to a recent technology is a technical skill and an approach with scientific accents is much more useful to the readers of the Electronics journal. I recommend major revision.
We have partially addressed this comment in our response to R2.1 and R 2.2. We note that running the CLAHE module in 4K is not only a matter of state-of-the-art technology, but also of redesigning known modules to work with the 4 ppc vector format (architectural contribution). We used the ZCU 104 platform, mainly because of the HDMI 2.0 I/O available on it, which allowed us to test the module in hardware. However, the module itself would easily work on previous generation chips - similar to the work of [6] and [8] (we also have some experience with a 4K signal for a Zynq SoC and Virtex 7 chip card).
However, in an attempt to address the comments as best we can, we have:
- added the discussion about real-time vision systems implemented in FPGA (Section 1 - Introduction)
- added Figure 1 - Comparison of Global Histogram Equalization, Adaptive Histogram Equalization and Contrast Limited Adaptive Histogram Equalization.
- added Figure 2 - Overview of the CLAHE algorithm.
- added discussion about applications of CLAHE (Section 2.5 - Applications of CLAHE) - mentioned in R2.6.
- added Figure 11 - Histogram calculation
- added Figure 12 - Histogram redistribution
- modified the description of Table 1 (to be more precise).
- added Figure 14 - Photo of the working system.
- added information about the maximal clock frequency (end of Section 5 - Results).
- added a whole new Section 6 - Discussion, where we present different possible variants of the CLAHE algorithm described in the literature (also for colour images) and comment on their hardware implementation in FPGA (all are possible with relatively minor modifications in the design).
We hope that in the current version the paper is more useful to the readers of the Electronics journal.
Tomasz Kryjak
Assistant professor
AGH University of Science and Technology in Kraków

Round 2
Reviewer 2 Report
In the new version of the paper is unrecognizable. I appreciate the effort of the authors, which is far beyond expectations. In this form, the work is extremely useful for readers of Electronics journal. I am sincerely impressed with the transformation of this work and I hope that the authors will be rewarded for this effort by many future citations.